# L2B: Learning to Bootstrap Robust Models for Combating Label Noise

## Abstract

Deep neural networks have shown great success in representation learning. However, when learning with noisy labels (LNL), they can easily overfit and fail to generalize to new data. To address this challenge, in this paper, we propose a novel machine learning method called Learning to Bootstrap (L2B) that leverages a joint reweighting mechanism to train models using their own predictions to bootstrap themselves without being adversely affected by erroneous pseudo-labels. Unlike conventional approaches, L2B dynamically adjusts the importance weight between real observed labels and pseudo-labels, as well as between different samples, to determine the appropriate weighting. Additionally, L2B conducts implicit relabeling concurrently, leading to significant improvements without incurring additional costs. L2B offers several benefits over the baseline methods. It yields more robust models that are less susceptible to the impact of noisy labels by guiding the bootstrapping procedure more effectively. It better exploits the valuable information contained in corrupted instances by adapting the weights of both instances and labels. Furthermore, L2B is compatible with existing noisy label learning methods and delivers competitive results on several benchmark datasets, including CIFAR-10, CIFAR-100, ISIC2019, and Clothing 1M datasets. Extensive experiments demonstrate that our method effectively mitigates the challenges of noisy labels, often necessitating few to no validation samples, and be well generalized to other tasks such as image segmentation. This not only positions it as a robust complement to existing LNL techniques but also underscores its practical applicability. The code and models are available at `https://anonymous.4open.science/r/L2B-6006`.

## 1 Introduction

In computer vision, deep learning has made significant strides, especially when provided with extensive, high-quality datasets. However, the persistent issue of label noise in real-world datasets, which stem from factors such as inter-observer variability, human annotation errors, and adversarial rival, can significantly undermine performance (Nettleton et al., 2010). As the size of datasets for deep learning continues to grow, the impact of label noise may become more significant. Understanding and addressing label noise is crucial for improving the accuracy and reliability of deep learning models (Liu et al., 2020; Wang et al., 2020; Zheng et al., 2021; Yao et al., 2021; Zhu et al., 2021; Wu et al., 2021b; Zhou et al., 2021).

Existing methods for learning with noisy labels (LNL) primarily focus on loss correction to counter noise effects. A common strategy is estimating the noise corruption matrix to adjust the loss function (Patrini et al., 2017; Goldberger & Ben-Reuven, 2017). However, correctly estimating the noise corruption matrix is usually challenging and often involves assumptions about the noise generation process (Xia et al., 2019; Liu & Tao, 2015; Hendrycks et al., 2018). Recent studies predominantly aim to identify and train on clean samples within noisy datasets (Jiang et al., 2018; Han et al., 2018; Yu et al., 2019), often considering low-loss samples as clean (Arpit et al., 2017). Rather than discarding noisy examples, meta-learning approaches have been to assign adaptive weights to each sample (Ren et al., 2018; Shu et al., 2019), with noisier samples given lower weights. However, this approach may compromise performance in high-noise scenarios by neglecting or underweighting portions of the training data.

To fully exploit the corrupted samples, a popular direction is to leverage the network predictions (i.e., pseudo-labels (Lee et al., 2013)) to recalibrate the labels (Reed et al., 2015; Tanaka et al., 2018; Song et al., 2019; Yi & Wu, 2019; Arazo et al., 2019). One representative work is the bootstrapping loss (Reed et al., 2015), which weights pseudo-labels in computing the training targets to counter the adverse effects of noisy samples. However the weight for pseudo-labels is often static, potentially leading to overfitting and poor label correction (Arazo et al., 2019). To tackle this challenge, Arazo et al. (2019) further designed a dynamic bootstrapping method, modulating the weight between actual and pseudo-labels by fitting a mixture model.

In contrast to prior works that individually reweight labels or instances, our paper introduces a novel approach to concurrently adjust both, elegantly unified under a meta-learning framework. We term our method as **L**earning to **B**ootstrap (**L2B**), as our goal is to enable the network to self-boost its capabilities by harnessing its own predictions in combating label noise. Specifcially, during each training iteration, L2B dynamically re-balances the importance between the true and pseudo labels as well as the per-sample weights, all of which are determined by the validation performance on a separated meta (clean) set in a meta-network. This differs from previous bootstrapping loss methods (Reed et al., 2015; Arazo et al., 2019; Zhang et al., 2020) that explicitly reassign labels using a weighted combination of pseudo and true labels. Importantly, unlike conventional reweighting mechanisms, L2B does not constrain these weights to sum to one. Furthermore, we empirically show that meta-learning algorithms' need for a clean validation set can be removed by dynamically creating an online meta set from the training data using a Gaussian mixture model (Permuter et al., 2006). This not only enhances our method's practicality but also facilitates its integration with current LNL techniques like DivideMix (Li et al., 2020), UniCon (Karim et al., 2022), and C2D (Zheltonozhskii et al., 2022). Consequently, L2B attains superior results without relying on a validation set.

In addition, we theoretically prove that our formulation, which reweights different loss terms, can be reduced to the original bootstrapping loss and therefore conducts an implicit relabeling instead. Through a meta-learning process, L2B achieves significant improvements (e.g., **+8.9%** improvement on CIFAR-100 with 50% noise) compared with the instance reweighting baseline with almost no extra cost. Our comprehensive tests across both natural and medical image datasets such as CIFAR-10, CIFAR-100, Clothing 1M, and ISIC2019, covering various types of label noise and recognition tasks, highlight L2B's superiority over contemporary label correction and meta-learning techniques.

## 2 RELATED WORKS

**Explicit relabeling.** Existing works propose to directly identify noisy samples and relabel them through estimating the noise transition matrix (Xia et al., 2019; Yao et al., 2020; Goldberger & Ben-Reuven, 2017; Patrini et al., 2017) or modeling noise by graph models or neural networks (Xiao et al., 2015; Vahdat, 2017; Veit et al., 2017; Lee et al., 2018). Patrini et al. (2017); Hendrycks et al. (2018) which estimate the label corruption matrix to directly correct the loss function. However, these methods usually require assumptions about noise modeling. For instance, Hendrycks et al. (2018) assume that the noisy label is only dependent on the true label and independent of the data. Another line of approaches proposes to leverage the network prediction (pseudo-labels) for explicit relabeling. Tanaka et al. (2018); Yi & Wu (2019) relabel the samples by directly using pseudo-labels in an iterative manner. Han et al. (2019) use generated prototypes as pseudo-labels to be more noise tolerant. Instead of directly using the pseudo-labels as supervision, Reed et al. (2015) propose to generate new training targets by a convex combination of the true and pseudo labels, furthered by Ortego et al. (2021a) for classification refinement. However, using a uniform weight for all samples, as in Reed et al. (2015), can exacerbate the influence of noisy data, impeding effective label correction. Semi-supervised LNL techniques like Li et al. (2020); Zhang et al. (2020) segment training data into labeled "clean samples" and unlabeled noisy sets, subsequently relabeled using pseudo-labels. To bolster the reliability of these pseudo-labels, unsupervised contrastive learning approaches are employed (Li et al., 2021; Ghosh & Lan, 2021; Zheltonozhskii et al., 2022; Karim et al., 2022).

**Instance reweighting.** To counteract the adverse effects of corrupted examples, various strategies focus on reweighting or selecting training instances to minimize the influence of noisy sam-

ples (Jiang et al., 2018; Ren et al., 2018; Fang et al., 2020). Based on the observation that deep neural networks tend to learn simple patterns first before fitting label noise (Arpit et al., 2017), many methods treat samples with small loss as clean ones (Jiang et al., 2018; Shen & Sanghavi, 2019; Han et al., 2018; Yu et al., 2019; Wei et al., 2020). Among those methods, Co-teaching (Han et al., 2018) and Co-teaching+ (Yu et al., 2019) train two networks to help select samples to train the other. Rather than directly selecting clean examples for training, meta-learning techniques (Ren et al., 2018; Shu et al., 2019; Xu et al., 2021) adjust instance weights, and curriculum learning (Jiang et al., 2018) sequences them by noise levels. Such strategies enhance robustness in medical imaging (Xue et al., 2019; Mirikharaji et al., 2019), but overlooking training subsets can affect performance in high-noise scenarios..

**Meta-learning.** Meta-learning based methods (Ren et al., 2018; Shu et al., 2019; Xu et al., 2021; Li et al., 2019; Wu et al., 2021a; Zheng et al., 2021; Zhang et al., 2020) aim to optimize model weights and hyper-parameters through a meta-process leveraging a small clean validation set. Among them, Ren et al. (2018); Shu et al. (2019); Xu et al. (2021) employ instance reweighting, adjusting example weights and network parameters through bi-level optimization to determine the contribution of each training sample. Wu et al. (2021a); Zheng et al. (2021); Zhang et al. (2020) approach label correction as a distinct meta-process.

Different from the aforementioned approaches which separately handle instance reweighting and label reweighting, we introduce a novel learning objective that concurrently meta-learns per-sample loss weights while implicitly relabeling the training data.

## 3 METHODOLOGY

### 3.1 PRELIMINARY

Given a set of $N$ training samples, i.e., $\mathcal{D}_{tra} = \{(x_i, y_i)|i = 1, ..., N\}$, where $x_i \in \mathbb{R}^{W \times H}$ denotes the $i$-th image and $y_i$ is the observed noisy label. In this work, we also assume that there is a small unbiased and clean validation set $\mathcal{D}_{val} = \{(x_i^v, y_i^v)|i = 1, ..., M\}$ and $M \ll N$, where the superscript $v$ denotes the validation set. Let $\mathcal{F}(:, \theta)$ denote the neural network model parameterized by $\theta$. Given an input-target pair $(x, y)$, we consider the loss function of $\mathcal{L}(\mathcal{F}(x, \theta), y)$ (e.g., cross-entropy loss) to minimize during the training process. Our goal, in this paper, is to properly utilize the small validation set $\mathcal{D}_{val}$ to guide the model training on $\mathcal{D}_{tra}$, for reducing the negative effects brought by the noisy annotation.

To establish a more robust training procedure, Reed et al. (2015) proposed the bootstrapping loss to enable the learner to "disagree" with the original training label, and effectively re-label the data during the training. Specifically, the training targets will be generated using a convex combination of training labels and predictions of the current model (i.e., pseudo-labels (Lee et al., 2013)), for purifying the training labels. Therefore, for a $L$-class classification problem, the loss function for optimizing $\theta$ can be derived as follows:

$$y_i^{\text{pseudo}} = \arg\max_{l=1,..,L} \mathcal{P}(x_i, \theta), \tag{1}$$

$$\theta^* = \arg\min_{\theta} \sum_{i=1}^{N} \mathcal{L}(\mathcal{F}(x_i, \theta), \beta y_i^{\text{real}} + (1 - \beta)y_i^{\text{pseudo}}), \tag{2}$$

where $\beta$ is used for balancing the weight between the real labels and the pseudo-labels. $\mathcal{P}(x_i, \theta)$ is the model output. $y^{\text{real}}$ and $y^{\text{pseudo}}$ denote the observed label and the pseudo-label respectively. However, in this method, $\beta$ is manually selected and fixed for all training samples, which does not prevent fitting the noisy ones and can even lead to low-quality label correction (Arazo et al., 2019). Moreover, we observe that this method is quite sensitive to the selection of the hyper-parameter $\beta$. For instance, as shown in Figure 1(a), even a similar $\beta$ selection (i.e., $\beta = 0.6$ vs. $\beta = 0.8$) behaves differently under disparate noise levels, making the selection of $\beta$ even more intractable. Another limitation lies in that Eq. 2 treats all examples as equally important during training, which could easily cause overfitting for biased training data.

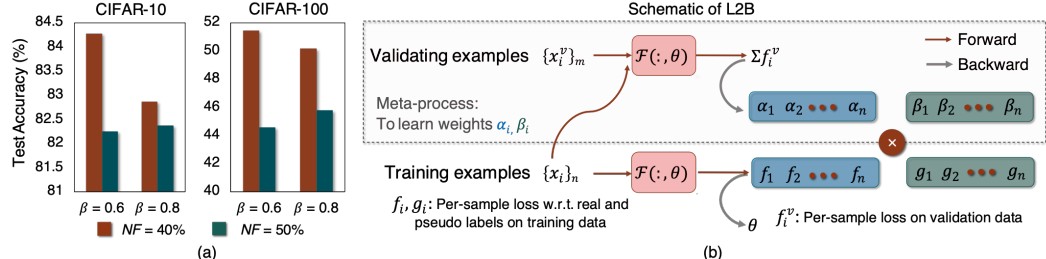

Figure 1: (a) The original bootstrapping loss (Reed et al., 2015) is sensitive to the reweighting hyper-parameter $\beta$. Under different noise levels, the optimal $\beta$ is different (*NF* stands for noise fraction). (b) Schematic description of our Learning to Bootstrap (i.e., **L2B**) method. The reweighting hyper-parameters are learned in a meta-process.

## 3.2 LEARNING TO BOOTSTRAP THROUGH META-LEARNING

To address these above challenges, in this paper, we aim to learn to bootstrap the model by conducting a joint label reweighting and instance reweighting. To achieve this, we propose to generate meta-learned weights for guiding our main learning objective:

$$\theta^*(\boldsymbol{\alpha}, \boldsymbol{\beta}) = \arg\min_{\theta} \sum_{i=1}^{N} \alpha_i \mathcal{L}(\mathcal{F}(x_i, \theta), y_i^{\text{real}}) + \beta_i \mathcal{L}(\mathcal{F}(x_i, \theta), y_i^{\text{pseudo}}), \tag{3}$$

with $\{\alpha_i, \beta_i\}_{i=1}^{N}$ being the balance weights. Here we note that this new learning objective can be regarded as a general form of the original bootstrapping loss, as Eq. 3 can be reduced to Eq. 2 when $\alpha_i + \beta_i = 1$ given that $\mathcal{L}(\cdot)$ is the cross-entropy loss (see details in Appendix B.1). By relaxing this constraint such that $\boldsymbol{\alpha}, \boldsymbol{\beta} \geq \mathbf{0}$, we can see that the optimization of Eq. 3 not only allows the main learner to explore the optimal combination between the two loss terms but also concurrently adjust the contribution of different training samples. In addition, compared with Eq. 2, the optimization of Eq. 3 does not rely on explicitly generating new training targets (i.e., $\beta y_i^{\text{real}} + (1 - \beta) y_i^{\text{pseudo}}$), but rather conducts implicit relabeling during training by reweighting different loss terms. We note that the key to L2B is that the sum of $\alpha_i$ and $\beta_i$ need not be 1, which results in **+8.9%** improvement on CIFAR-100 with 50% noise (Section 3.5).

Note that this form is also similar to self-distillation in Li et al. (2017). But different from Li et al. (2017) where the weights are determined by heuristics, our weights $\boldsymbol{\alpha}, \boldsymbol{\beta}$ are meta-learned based on its performance on the validation set $\mathcal{D}_{val}$, that is

$$\boldsymbol{\alpha}^*, \boldsymbol{\beta}^* = \arg\min_{\boldsymbol{\alpha}, \boldsymbol{\beta} \geq \mathbf{0}} \frac{1}{M} \sum_{i=1}^{M} \mathcal{L}(\mathcal{F}(x_i^v, \theta^*(\boldsymbol{\alpha}, \boldsymbol{\beta})), y_i^v). \tag{4}$$

It is necessary to constrain $\alpha_i, \beta_i \geq 0$ for all $i$ to avoid potential unstable training (Ren et al., 2018). Both the meta learner (i.e., Eq. 4) and the main learner (i.e., Eq. 3) are optimized concurrently, which allows the model to maximize the performance on the clean validation set $\mathcal{D}_{val}$ by adjusting the importance weights of the observed and the pseudo-labels in a differentiable manner.

**Online Approximation.** For each step $t$ at training, a mini-batch of training examples $\{(x_i, y_i), 1 \leq i \leq n\}$ with $n \ll N$ is sampled to estimate a temporary adjustment to the parameters based on the descent direction of the loss function. For simplicity, let $f_i(\theta)$ denote $\mathcal{L}(\mathcal{F}(x_i, \theta), y_i^{\text{real}})$ and $g_i(\theta)$ denote $\mathcal{L}(\mathcal{F}(x_i, \theta), y_i^{\text{pseudo}})$ in the following sections. Given any $\boldsymbol{\alpha}, \boldsymbol{\beta}$, we use

$$\hat{\theta}_{t+1} = \theta_t - \lambda \nabla \left( \sum_{i=1}^{n} \alpha_i \, f_i(\theta) + \beta_i \, g_i(\theta) \right) \Big|_{\theta=\theta_t} \tag{5}$$

to approach the solution of Eq. 3. Here $\lambda$ is the step size. We then estimate the corresponding optimal $\boldsymbol{\alpha}, \boldsymbol{\beta}$ as

$$\boldsymbol{\alpha}_t^*, \boldsymbol{\beta}_t^* = \arg\min_{\boldsymbol{\alpha}, \boldsymbol{\beta} \geq \mathbf{0}} \frac{1}{M} \sum_{i=1}^{M} f_i^v(\hat{\theta}_{t+1}). \tag{6}$$

---

**Algorithm 1** Learning to Bootstrap

---

**Require:** $\theta_0, \mathcal{D}_{tra}, \mathcal{D}_{val}, n, m, L$
**Ensure:** $\theta_T$
 1: **for** $t = 0 \ ... \ T - 1$ **do**
 2:     $\{x_i, y_i\} \leftarrow \text{SampleMiniBatch}(\mathcal{D}_{tra}, n)$
 3:     $\{x_i^v, y_i^v\} \leftarrow \text{SampleMiniBatch}(\mathcal{D}_{val}, m)$
 4:     For the $i$-th sample of $\mathcal{D}_{tra}$, compute $y_i^{\text{pseudo}} = \arg\max_{l=1,..,L} \mathcal{P}(x_i, \theta_t)$
 5:     Learnable weights $\boldsymbol{\alpha}, \boldsymbol{\beta}$
 6:     Compute training loss $l_f \leftarrow \sum_{i=1}^n \alpha_i f_i(\theta_t) + \beta_i g_i(\theta_t)$
 7:     $\hat{\theta}_{t+1} \leftarrow \theta_t - \lambda \nabla l_f \big|_{\theta=\theta_t}$
 8:     Compute validation loss $l_g \leftarrow \frac{1}{m} \sum_{i=1}^m f_i^v(\hat{\theta}_{t+1})$
 9:     $(\boldsymbol{\alpha}_t, \boldsymbol{\beta}_t) \leftarrow -\eta \nabla l_g \big|_{\boldsymbol{\alpha}=0, \boldsymbol{\beta}=0}$
10:     $\tilde{\alpha}_{t,i} \leftarrow \max(\alpha_{t,i}, 0), \ \tilde{\beta}_{t,i} \leftarrow \max(\beta_{t,i}, 0)$
11:     $\tilde{\alpha}_{t,i} \leftarrow \frac{\tilde{\alpha}_{t,i}}{\sum_{i=1}^n \tilde{\alpha}_{t,i} + \tilde{\beta}_{t,i}}, \ \tilde{\beta}_{t,i} \leftarrow \frac{\tilde{\beta}_{t,i}}{\sum_{i=1}^n \tilde{\alpha}_{t,i} + \tilde{\beta}_{t,i}}$
12:     Apply learned weights $\boldsymbol{\alpha}, \boldsymbol{\beta}$ to reweight the training loss as $\hat{l}_f \leftarrow \sum_{i=1}^n \tilde{\alpha}_{t,i} f_i(\theta_t) + \tilde{\beta}_{t,i} g_i(\theta_t)$
13:     $\theta_{t+1} \leftarrow \theta_t - \lambda \nabla \hat{l}_f \big|_{\theta=\theta_t}$
14: **end for**

---

However, directly solving for Eq. 6 at every training step requires too much computation cost. To reduce the computational complexity, we apply one step gradient descent of $\boldsymbol{\alpha}_t, \boldsymbol{\beta}_t$ on a mini-batch of validation set $\{(x_i^v, y_i^v), 1 \le i \le m\}$ with $m \le M$ as an approximation. Specifically,

$$(\alpha_{t,i}, \beta_{t,i}) = -\eta \nabla \Big( \sum_{i=1}^m f_i^v(\hat{\theta}_{t+1}) \Big) \Big|_{\alpha_i=0, \beta_i=0}, \tag{7}$$

where $\eta$ is the step size for updating $\boldsymbol{\alpha}, \boldsymbol{\beta}$. To ensure that the weights are non-negative, we apply the following rectified function:

$$\tilde{\alpha}_{t,i} = \max(\alpha_{t,i}, 0), \ \tilde{\beta}_{t,i} = \max(\beta_{t,i}, 0). \tag{8}$$

To stabilize the training process, we also normalize the weights in a single training batch so that they sum up to one:

$$\tilde{\alpha}_{t,i} = \frac{\tilde{\alpha}_{t,i}}{\sum_{i=1}^n \tilde{\alpha}_{t,i} + \tilde{\beta}_{t,i}}, \ \tilde{\beta}_{t,i} = \frac{\tilde{\beta}_{t,i}}{\sum_{i=1}^n \tilde{\alpha}_{t,i} + \tilde{\beta}_{t,i}}. \tag{9}$$

Finally, we estimate $\theta_{t+1}$ based on the updated $\boldsymbol{\alpha}_t, \boldsymbol{\beta}_t$ so that $\theta_{t+1}$ can consider the meta information included in $\boldsymbol{\alpha}_t, \boldsymbol{\beta}_t$:

$$\theta_{t+1} = \theta_t - \lambda \nabla \Big( \sum_{i=1}^n \tilde{\alpha}_{t,i} \, f_i(\theta) + \tilde{\beta}_{t,i} \, g_i(\theta) \Big) \Big|_{\theta=\theta_t}. \tag{10}$$

See Appendix B.2 for detailed calculation of the gradient in Eq. 10. A schematic description of our Learning to Bootstrap algorithm is illustrated in Figure 1(b) and the overall optimization procedure can be found in Algorithm 1.

### 3.3 CONVERGENCE ANALYSIS

In proposing Eq. 3, we show that with the first-order approximation of $\boldsymbol{\alpha}, \boldsymbol{\beta}$ in Eq. 7 and some mild assumptions, our method guarantees to convergence to a local minimum point of the validation loss, which yields the best combination of $\boldsymbol{\alpha}, \boldsymbol{\beta}$. Details of the proof are provided in Appendix B.3.

**Theorem 1.** *Suppose that the training loss function $f, g$ have $\sigma$-bounded gradients and the validation loss $f^v$ is Lipschitz smooth with constant L. With a small enough learning rate $\lambda$, the validation loss monotonically decreases for any training batch B, namely,*

$$G(\theta_{t+1}) \le G(\theta_t), \tag{11}$$

*where $\theta_{t+1}$ is obtained using Eq. 10 and G is the validation loss*

$$G(\theta) = \frac{1}{M} \sum_{i=1}^{M} f_i^v(\theta),$$ (12)

*Furthermore, Eq. 11 holds for all possible training batches only when the gradient of validation loss function becomes 0 at some step t, namely, $G(\theta_{t+1}) = G(\theta_t) \; \forall B \Leftrightarrow \nabla G(\theta_t) = 0$*

## 3.4 DATASETS

**CIFAR-10 & CIFAR-100.** Both CIFAR-10 and CIFAR-100 contain 50K training images and 10K test images of size $32 \times 32$. Following previous works (Tanaka et al., 2018; Kim et al., 2019; Li et al., 2020), we experimented with both *symmetric* and *asymmetric* label noise. In our method, we used 1,000 clean images in the validation set $\mathcal{D}_{val}$ following Jiang et al. (2018); Ren et al. (2018); Shu et al. (2019); Hendrycks et al. (2018); Zheng et al. (2021).

**ISIC2019.** Following Xue et al. (2019), we also evaluated our algorithm on a medical image dataset, i.e., skin lesion classification data, under different symmetric noise levels. Our experiments were conducted on the 25,331 dermoscopic images of the 2019 ISIC Challenge[1], where we used 20400 images as the training set $\mathcal{D}_{tra}$, 640 images as the validation set $\mathcal{D}_{val}$, and tested on 4291 images.

**Clothing 1M.** We evaluate on real-world noisy dataset, Clothing 1M (Xiao et al., 2015), which has 1 million training images collected from online shopping websites with labels generated from surrounding texts. In addition, the Clothing 1M also provides an official validation set of 14,313 images and a test set of 10,526 images. Implementation details can be found in Appendix A.1.

## 3.5 PERFORMANCE COMPARISONS

**Efficacy of L2B.** We compare our method with different baselines: 1) Cross-Entropy (the standard training), 2) Bootstrap Reed et al. (2015), which modifies the training loss by generating new training targets , and 3) L2RW (Ren et al., 2018), which reweights different instances through meta-learning under different levels of symmetric labels noise ranging from $20\% \sim 50\%$. To ensure a fair comparison, we report the best epoch for all comparison approaches. All results are summarized in Table 1. Compared with the naive bootstrap method and the baseline meta-learning-based instance reweighting method L2RW, the performance improvement is substantial, especially under larger noise fraction, which suggests that using meta-learning to automatically bootstrap the model is more beneficial for LNL. For example, on CIFAR-100, the accuracy improvement of our proposed L2B reaches $7.6\%$ and $8.9\%$ under $40\%$ and $50\%$ noise fraction, respectively. We also show a set qualitative examples to illustrate how L2B adjust the weights to rectify the influence from the noisy labels in Figure 4.

**Comparison with the state-of-the-arts.** We compare our method with SOTA methods on CIFAR 10 and CIFAR 100 in Table 2. We demonstrate our L2B is compatible with existing LNL methods. When integrated with existing LNL methods like DivideMix (Li et al., 2020), UniCon (Karim et al., 2022), C2D (Zheltonozhskii et al., 2022), L2B consistently enhances performance across varying noise ratios on both datasets. Notably, L2B-C2D surpasses all competing methods in various settings, achieving $94.4\%$ and $60.7\%$ accuracy under the noise ratio of $90\%$ for CIFAR-10 and CIFAR-100. We also test our model with $40\%$ asymmetric noise and summarize the testing accuracy in Table 3. Among all compared methods, we re-implement L2RW under the same setting and report the performance of all other competitors from previous papers including Kim et al. (2019; 2021); Li et al. (2020).Compared with previous meta-learning-based methods (*e.g.,* Chen et al. (2019), Zhang & Yao (2020)), and other methods (*e.g.,* Ren et al. (2018), Wu et al. (2021a), Shu et al. (2019)), our L2B achieves superior results.

---

[1] https://challenge2019.isic-archive.com/data.html

Table 1: Comparison in test accuracy (%) with the baseline methods on CIFAR-10/100 datasets with symmetric noise.

| Dataset | CIFAR-10 | | | | CIFAR-100 | | | | ISIC | | | |
|---|---|---|---|---|---|---|---|---|---|---|---|---|
| Method/Noise ratio | 20% | 30% | 40% | 50% | 20% | 30% | 40% | 50% | 20% | 30% | 40% | 50% |
| Cross-Entropy (CE) | 86.9 | 84.9 | 83.3 | 81.3 | 59.6 | 52.2 | 49.2 | 44.4 | 79.4 | 77.5 | 75.3 | 73.7 |
| Bootstrap (Reed et al., 2015) | 85.2 | 84.8 | 82.9 | 79.2 | 61.8 | 54.2 | 50.2 | 45.8 | 80.8 | 77.7 | 75.7 | 74.8 |
| L2RW (Ren et al., 2018) | 90.6 | 89.0 | 86.6 | 85.3 | 67.8 | 63.8 | 59.7 | 55.6 | 80.1 | 77.7 | 76.3 | 74.1 |
| L2B (Ours) | **92.2** | **90.7** | **89.9** | **88.5** | **71.8** | **69.5** | **67.3** | **64.5** | **81.1** | **80.2** | **78.6** | **76.8** |

Table 2: Comparison in test accuracy (%) with state-of-the-art methods on CIFAR-10/100 datasets with symmetric noise.

| Dataset | CIFAR-10 | | | | CIFAR-100 | | | |
|---|---|---|---|---|---|---|---|---|
| Method/Noise ratio | 20% | 50% | 80% | 90% | 20% | 50% | 80% | 90% |
| Co-teaching+ (Yu et al., 2019) | 89.5 | 85.7 | 67.4 | 47.9 | 65.6 | 51.8 | 27.9 | 13.7 |
| Mixup (Zhang et al., 2018) | 95.6 | 87.1 | 71.6 | 52.2 | 67.8 | 57.3 | 30.8 | 14.6 |
| PENCIL (Yi & Wu, 2019) | 92.4 | 89.1 | 77.5 | 58.9 | 69.4 | 57.5 | 31.1 | 15.3 |
| Meta-Learning (Li et al., 2019) | 92.9 | 89.3 | 77.4 | 58.7 | 68.5 | 59.2 | 42.4 | 19.5 |
| M-correction (Arazo et al., 2019) | 94.0 | 92.0 | 86.8 | 69.1 | 73.9 | 66.1 | 48.2 | 24.3 |
| AugDesc (Nishi et al., 2021) | 96.3 | 95.4 | 93.8 | 91.9 | 79.5 | 77.2 | 66.4 | 41.2 |
| GCE (Ghosh & Lan, 2021) | 90.0 | 89.3 | 73.9 | 36.5 | 68.1 | 53.3 | 22.1 | 8.9 |
| Sel-CL+ (Li et al., 2022) | 95.5 | 93.9 | 89.2 | 81.9 | 76.5 | 72.4 | 59.6 | 48.8 |
| MLC (Zheng et al., 2021) | 92.6 | 88.1 | 77.4 | 67.9 | 66.8 | 52.7 | 21.8 | 15.0 |
| MSLC (Wu et al., 2021a) | 93.4 | 89.9 | 69.8 | 56.1 | 72.5 | 65.4 | 24.3 | 16.7 |
| MOIT+ (Ortego et al., 2021b) | 94.1 | 91.8 | 81.1 | 74.7 | 75.9 | 70.6 | 47.6 | 41.8 |
| DivideMix (Li et al., 2020) | 96.1 | 94.6 | 93.2 | 76.0 | 77.3 | 74.6 | 60.2 | 31.5 |
| L2B-DivideMix | 96.1 | **95.4** | **94.0** | **91.3** | **77.9** | **75.9** | **62.2** | **35.8** |
| UniCon (Karim et al., 2022) | 96.0 | 95.6 | 93.9 | 90.8 | 78.9 | 77.6 | 63.9 | 44.8 |
| L2B-UniCon | **96.5** | **95.8** | **94.7** | **92.8** | 78.8 | 77.3 | **67.6** | **49.6** |
| C2D (Zheltonozhskii et al., 2022) | 96.3 | 95.2 | 94.4 | 93.5 | 78.7 | 76.4 | 67.8 | 58.7 |
| L2B-C2D | **96.7** | **95.6** | **94.8** | **94.4** | **80.1** | **78.1** | **69.6** | **60.7** |

Table 3: Comparison with 40% asymmetric noise in test accuracy on the CIFAR-10 dataset.

| Method | Acc |
|---|---|
| Cross-Entropy | 85.0 |
| F-correction (Patrini et al., 2017) | 87.2 |
| M-correction (Arazo et al., 2019) | 87.4 |
| Chen et al. (Chen et al., 2019) | 88.6 |
| P-correction (Yi & Wu, 2019) | 88.5 |
| REED (Zhang & Yao, 2020) | 92.3 |
| Tanaka et al. (Tanaka et al., 2018) | 88.9 |
| NLNL (Kim et al., 2019) | 89.9 |
| JNPL (Kim et al., 2021) | 90.7 |
| DivideMix (Li et al., 2020) | 93.4 |
| MLNT (Li et al., 2019) | 89.2 |
| L2RW (Ren et al., 2018) | 89.2 |
| MW-Net (Shu et al., 2019) | 89.7 |
| MSLC (Wu et al., 2021a) | 91.6 |
| Meta-Learning (Li et al., 2019) | 88.6 |
| Distilling (Zhang et al., 2020) | 90.2 |
| L2B-Naive (Ours) | 91.8 |
| L2B-C2D (Ours) | **94.0** |

Table 4: Comparison with state-of-the-art methods in test accuracy (%) on Clothing 1M.

| Method | Acc |
|---|---|
| CrossEntropy | 69.2 |
| M-correction (Arazo et al., 2019) | 71.0 |
| PENCIL (Yi & Wu, 2019) | 73.5 |
| DivideMix (Li et al., 2020) | 74.8 |
| Nested (Chen et al., 2021) | 74.9 |
| AugDesc (Nishi et al., 2021) | 75.1 |
| RRL (Li et al., 2021) | 74.9 |
| GCE (Ghosh & Lan, 2021) | 73.3 |
| C2D (Zheltonozhskii et al., 2022) | 74.3 |
| MLNT (Li et al., 2019) | 73.5 |
| MLC (Zheng et al., 2021) | 75.8 |
| MSLC (Wu et al., 2021a) | 74.0 |
| Meta-Cleaner (Zhang et al., 2019) | 72.5 |
| Meta-Weight (Shu et al., 2019) | 73.7 |
| FaMUS (Xu et al., 2021) | 74.4 |
| MSLG (Algan & Ulusoy, 2021) | 76.0 |
| L2B-Naive (Ours) | **77.5 ± 0.2** |

**Generalization to real-world noisy labels.** We test L2B on Clothing 1M (Xiao et al., 2015), a large-scale dataset with real-world noisy labels. The results of all competitors are reported from published papers. As shown in Table 4, our L2B-Naive attains an average performance of 77.5% accuracy from 3 independent runs with different random seeds, outperforming all competing methods.

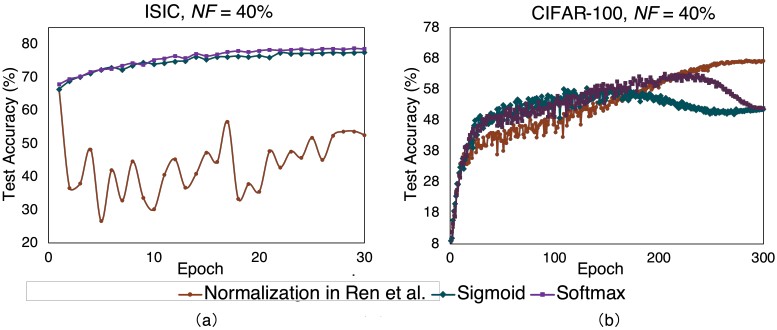

Figure 2: Comparison among different normalization functions (i.e., Eq. 9, Sigmoid function and Softmax function). Testing accuracy curve: (a) with different normalization functions under 40% symmetric noise label on the ISIC dataset. (b) with different normalization under 40% symmetric label noise on CIFAR-100.

**Generalization to image segmentation** L2B can be easily generalized for segmentation tasks. Specifically, the learnable weights $\alpha$ and $\beta$ are replaced with pixel-wise weight maps corresponding to noisy labels and pseudo labels (model predictions). L2B dynamically assigns these weight maps, adjusting for both noisy and pseudo labels to optimize the bootstrapping process via a meta-process. To assess L2B's performance in segmentation, we employed the PROMISE12 dataset Litjens et al. (2014) which contains 50 3D transversal T2-weighted MR images. Specifically, 40/10 cases were used for training/evaluation. 3 out of the 40 training cases are chosen randomly as the meta set. As presented in Table 5, we compare our method with 1) UNet++ (Zhou et al., 2018), 2) UNet++ meta, which trains exclusively on the meta data, 3) NL reweighting (Mirikharaji et al., 2019), which only reweights the noisy labels, 4) Mix-up (Zhang et al., 2017), a regularization based method. L2B outperforms others in all evaluation metrics of Dice, Jaccard Index (JI), Hausdorff Distance (HD) and Average Surface Distance (ASD). More detailed analysis could be found in Appendix A.4.

### 3.6 ABLATION STUDY

**On the importance of $\alpha, \beta$.** To understand why our proposed new learning objective can outperform previous meta-learning-based instance reweighting methods, we conduct the following analysis to understand the importance of hyper-parameter $\alpha$ and $\beta$ in our method. Specifically, we set $\alpha = 0$ and $\beta = 0$ respectively to investigate the importance of each loss term in Eq. equation 3. In addition, we also show how the restriction of $\alpha_i + \beta_i = 1$ (Eq. equation 2) would deteriorate our model performance as follows.

- $\alpha = 0$. As shown in Table 6, in this case, the performance even decreases compared with the baseline approach. This is due to that when only pseudo-labels are included in the loss computation, the error which occurs in the initial pseudo-label will be reinforced by the network during the following iterations.
- $\beta = 0$. From Eq. equation 3, we can see that setting $\beta$ as $0$ is essentially equivalent to the baseline meta-learning-based instance reweighting method L2RW (Ren et al., 2018). In this case, the performance is largely improved compared to the baseline, but still inferior to our method, which jointly optimizes $\alpha$ and $\beta$.
- $\alpha + \beta = 1$. We also investigate whether the restriction of $\alpha + \beta = 1$ is required for obtaining optimal weights during the meta-update, as in (Zhang et al., 2020). As shown in Table 6, L2B ($\alpha, \beta \geq 0$) consistently achieves superior results than L2B ($\alpha + \beta = 1$) under different noise levels on CIFAR-100. The reason may be the latter is only reweighting different loss terms, whereas the former not only explores the optimal combination between the two loss terms but also jointly adjusts the contribution of different training samples.

**Parameter normalization** We note that the normalization of $\alpha$ and $\beta$ is one key component for accelerating the training process. However, we observe that different normalization methods of $\alpha$ and $\beta$ behave quite differently for different datasets. To further investigate this, we apply the following normalization functions to each $\alpha_i$ and $\beta_i$ on ISIC2019, CIFAR-100, and Clothing 1M: 1) Eq. 9 as in (Ren et al., 2018), 2) Sigmoid function,

$$\alpha_{t,i} = \frac{1}{1 + e^{-\alpha_{t,i}}}, \; \beta_{t,i} = \frac{1}{1 + e^{-\beta_{t,i}}}, \tag{13}$$

and 3) Softmax function,

$$\alpha_{t,i} = \frac{e^{\alpha_{t,i}/\tau}}{\sum_{i=1}^{n} e^{\alpha_{t,i}/\tau} + e^{\beta_{t,i}/\tau}}, \ \beta_{t,i} = \frac{e^{\beta_{t,i}/\tau}}{\sum_{i=1}^{n} e^{\alpha_{t,i}/\tau} + e^{\beta_{t,i}/\tau}}, \tag{14}$$

where $t$ stands for the training iteration and $\tau$ denotes the temperature parameter for scaling the weight distribution. $\tau$ is set as 10.0 when using the Softmax function for normalization. The comparison among these three different normalization methods is summarized in Figure 2 on ISIC2019 and CIFAR-100 datasets with 40% symmetric noise. We can see that while Eq. 9 achieves the best result on CIFAR-100, it yields large training instability on the ISIC2019 dataset. Changing the normalization function to Sigmoid and Softmax can make the training procedure much more stable on the ISIC2019 dataset.

Table 5: Performance comparison under noisy-supervision on PROMISE12.

| Method | Dice (%)↑ | JI (%)↑ | HD (voxel)↓ | ASD (voxel)↓ |
|---|---|---|---|---|
| UNet++ Zhou et al. (2018) | 73.74 | 58.90 | 11.63 | 3.70 |
| UNet++ meta | 73.04 | 58.51 | 17.06 | 5.50 |
| NL reweighting Mirikharaji et al. (2019) | 76.64 | 62.62 | 8.33 | 2.75 |
| Mix-up Zhang et al. (2017) | 69.18 | 53.78 | 13.25 | 4.56 |
| L2B (Ours) | **80.83** | **68.10** | **6.68** | **2.10** |

Table 6: Ablation of $\alpha, \beta$.

| Method | 20% | 40% |
|---|---|---|
| baseline (CE) | 59.6 | 49.2 |
| $\alpha = 0$ | 55.7 | 47.1 |
| $\beta = 0$ | 63.2 | 57.5 |
| $\alpha + \beta = 1$ | 64.8 | 59.1 |
| $\alpha, \beta \geq 0$ | **71.8** | **67.3** |

Table 7: Ablation on size of validation data on CIFAR-10 and CIFAR-100 datasets.

| | | CIFAR-10 | | | | CIFAR-100 | | | |
|---|---|---|---|---|---|---|---|---|---|
| | Validation Size | 20% | 50% | 80% | 90% | 20% | 50% | 80% | 90% |
| L2B-DivideMix | baseline | 96.1 | 94.6 | 93.2 | 76.0 | 77.3 | 74.6 | 60.2 | 31.5 |
| | 0 | 96.3 | 95.3 | 93.5 | 82.6 | 77.6 | 75.3 | 60.8 | 31.0 |
| | 500 | 96.1 | 95.3 | 93.8 | 91.1 | 78.2 | 75.3 | 62.5 | 34.0 |
| | 1000 | 96.1 | 95.4 | 94.0 | 91.3 | 77.9 | 75.9 | 62.2 | 35.8 |
| L2B-UniCon | baseline | 96.0 | 95.6 | 93.9 | 90.8 | 78.9 | 77.6 | 63.9 | 44.8 |
| | 0 | 96.4 | 95.6 | 94.2 | 92.5 | 78.7 | 77.4 | 68.0 | 48.6 |
| | 500 | 96.3 | 95.6 | 94.5 | 92.7 | 78.5 | 77.5 | 67.8 | 51.1 |
| | 1000 | 96.5 | 95.8 | 94.7 | 92.8 | 78.8 | 77.3 | 67.6 | 49.6 |
| L2B-C2D | baseline | 96.4 | 95.3 | 94.4 | 93.5 | 78.7 | 76.4 | 67.8 | 58.7 |
| | 0 | 96.4 | 95.6 | 94.9 | 93.7 | 79.1 | 77.8 | 68.5 | 60.3 |
| | 500 | 96.6 | 95.5 | 94.9 | 94.0 | 79.5 | 77.9 | 69.0 | 60.8 |
| | 1000 | 96.7 | 95.6 | 94.8 | 94.4 | 80.1 | 78.1 | 69.6 | 60.7 |

**The number of clean validation samples** In Table 7, our L2B method is shown to require few to no validation samples for LNL problems, highlighting its practicality. L2B consistently boosts baseline methods such as DivideMix, UniCon, and C2D. Specifically, L2B-DivideMix has showcased its efficacy, particularly at high noise levels. Specifically, in a scenario with 90% noise on CIFAR-10, our approach outstripped the baseline by 8.7%, achieving an accuracy of 82.6% compared to 76.0%, and this was achieved without the need for clean validation samples. The advantage of L2B-DivideMix becomes even more pronounced when we incorporate a minimal amount of clean labels. With just 500 clean labels (equivalent to 2% of the training data), our performance lead over the baseline extends to a remarkable 15.1%. However, as we double the clean samples to 1000, the incremental benefit tapers off, yielding a mere 0.2% boost. This behavior underscores the efficiency of L2B-DivideMix, demonstrating that it can deliver impressive results with minimal or even no clean validation data, making it a highly adaptable and practical solution for real-world applications.

## 4 CONCLUSION

Our paper presents Learning to Bootstrap (L2B), a new technique using joint reweighting for model training. L2B dynamically balances weights between actual labels, pseudo-labels, and different samples, mitigating the challenges of erroneous pseudo-labels. Notably, L2B operates effectively without a clean validation set and can be well generalized to other tasks, highlighting its practicality in real-world settings. Extensive experiments on CIFAR-10, CIFAR-100, ISIC2019, and Clothing 1M datasets demonstrate the superiority and robustness compared to other existing methods under various settings.

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

## A Appendix

### A.1 Implementation Details

For all CIFAR-10 and CIFAR-100 comparison experiments, we used an 18-layer PreActResNet (He et al., 2016) as the baseline network following the setups in (Li et al., 2020), unless otherwise specified. The model was trained using SGD with a momentum of 0.9, a weight decay of 0.0005, and a batch size of 256 for CIFAR-100 and 512 for CIFAR-10. The network was trained from scratch for 300 epochs. We set the learning rate as 0.15 initially with a cosine annealing decay. Following (Li et al., 2020), we set the warm up period as 10 epochs for both CIFAR-10 & CIFAR-100. The optimizer and the learning rate schedule remained the same for both the main and the meta model. Gradient clipping is applied to stabilize training. All experiments were conducted with one V100 GPU, except for the experiments on Clothing 1M which were conducted with one RTX A6000 GPU.

For ISIC2019 experiments, we used ResNet-50 with ImageNet pretrained weights. A batch size of 64 was used for training with an initial learning rate of 0.01. The network was trained for 30 epochs in total with the warmup period as 1 epoch. All other implementation details remained the same as above. For Clothing 1M experiments, we used an ImageNet pre-trained 18-layer ResNet (He et al., 2016) as our baseline. We finetuned the network with a learning rate of 0.005 for 300 epochs. The model was trained using SGD with a momentum of 0.9, a weight decay of 0.0005, and a batch size of 256. Following (Li et al., 2020), to ensure the labels (noisy) were balanced, for each epoch, we sampled 250 mini-batches from the training data.

### A.2 Alleviate potential overfitting to noisy examples.

We also plot the testing accuracy curve under different noise fractions in Figure 3, which shows that our proposed L2B would help preventing potential overfitting to noisy samples compared with standard training. Meanwhile, compared to simply sample reweighting (L2RW), our L2B introduces pseudo-labels for bootstrapping the learner and is able to converge to a better optimum.

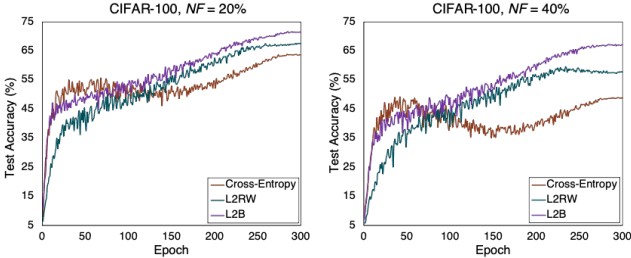

Figure 3: Test accuracy v.s. number of epochs on CIFAR-100 under the noise fraction of 20% and 40%.

### A.3 Qualitative Results

We also demonstrate a set of qualitative examples to illustrate how our proposed L2B benefits from the joint instance and label reweighting paradigm. In Figure 4, we can see that when the estimated pseudo label is of high-quality, i.e., the pseudo label is different from the noisy label but equal to the clean label, our model will automatically assign a much higher weight to $\beta$ for corrupted training samples. On the contrary, $\alpha$ can be near zero in this case. This indicates that our L2B algorithm will pay more attention to the pseudo label than the real noisy label when computing the losses. In addition, we also show several cases where the pseudo label is equal to the noisy label, where we can see that $\alpha$ and $\beta$ are almost identical under this circumstance since the two losses are of the same value. Note that the relatively small values of $\alpha$ and $\beta$ are due to that we use a large batch size (i.e., 512) for CIFAR-10 experiments. By normalizing the weights in each training batch (see Eq. 9), the value of $\alpha$ and $\beta$ can be on the scale of $10^{-4}$.

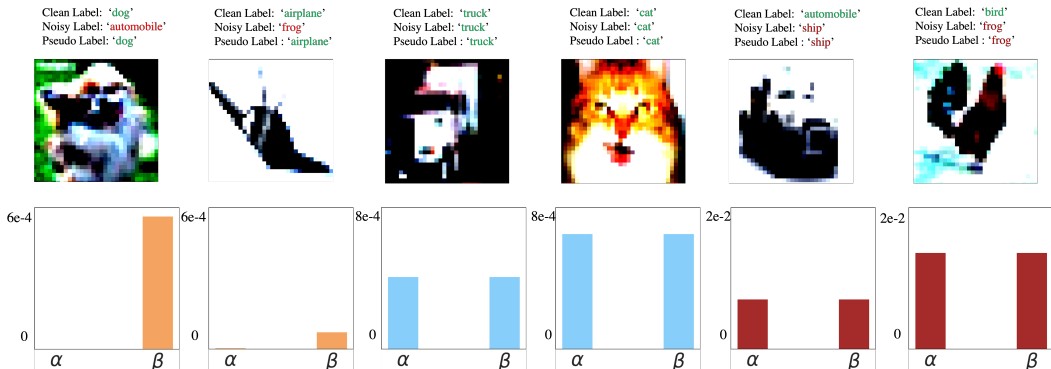

Figure 4: Examples of $\alpha$ and $\beta$ on CIFAR-10 with asymmetric noise fraction of 20%. When the estimated pseudo label is of high-quality, i.e., the pseudo label is different from the noisy label but equal to the clean label, our model will automatically assign a much higher weight to $\beta$ than to $\alpha$ for corrupted training samples. When the pseudo label is equal to the noisy label (i.e., the two loss terms are equal to each other), $\alpha$ and $\beta$ are almost identical.

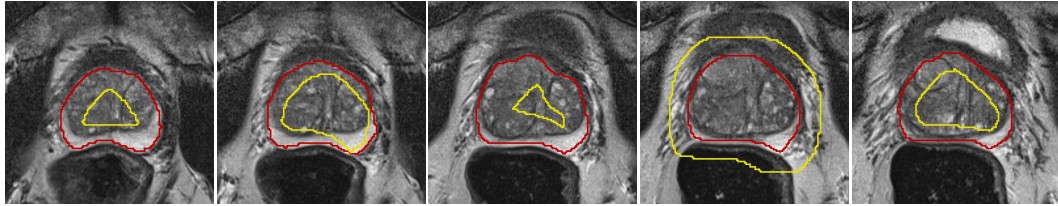

Figure 5: Visual comparison of prostate MRI images with noisy (contoured in yellow) and accurate (contoured in red) segmentation masks to demonstrate the discrepancy in segmentation quality between the two.

### A.4 DETAILS OF THE GENERALIZED SEGMENTATION TASK

PROMISE12 dataset contains 50 3D transversal T2-weighted MR images of the prostate with manual binary prostate gland segmentation and is obtained from multiple centers with different acquisition protocols. Following Soerensen et al. (2021); Wang et al. (2021), we utilized 2D slices in the axial view for both training and testing. All images are resized to $144 \times 144$ and splits are randomized. Noisy labels used in Table 5 were synthesized using random rotation, erosion, or dilation, achieving approximately a 60% corruption ratio and an average Dice coefficient of $0.6206$. And visualizations of the corrupted noisy labels (shown in yellow) as well as the ground-truth (shown in red) are illustrated in Figure 5. Furthermore, we also investigate the robustness of our method by varying the noise level of the corrupted training set from $\{L_1, L_2, L_3\}$, where the average Dice coefficients are $\text{Dice}_{L_1} = 0.4148$, $\text{Dice}_{L_2} = 0.6206$, and $\text{Dice}_{L_3} = 0.8031$ (*i.e.*, the corrupted ratios are around 60% ($L_1$), 40% ($L_2$), and 20% ($L_3$)). At each noise level, we compare the baseline UNet++ which is directly trained on the noisy training data with our MLB-Seg. As shown in Table 8, we report the averaged dice coefficient over 5 repetitions for each series of experiments. The standard deviation for all experiments is within 0.5%. We could notice that while the noise level increases, performances of baseline drop from 80.03% to 59.77%, but performances of MLB-Seg only drop

Table 8: Ablation study on different noise levels

| Method | Dice (%)↑ |
|---|---|
| baseline - $L_1$ | 59.77 |
| **MLB-Seg** - $L_1$ | 77.70 |
| baseline - $L_2$ | 73.74 |
| **MLB-Seg** - $L_2$ | 80.83 |
| baseline - $L_3$ | 80.03 |
| **MLB-Seg** - $L_3$ | 82.01 |

from 82.01% to 77.70% which indicates that our MLB-Seg is robust to different noisy levels and shows larger improvements under a much severer noisy situation.

## B THEORETICAL ANALYSIS

### B.1 EQUIVALENCE OF THE TWO LEARNING OBJECTIVES

We show that Eq. 3 is equivalent with Eq. 2 when $\forall i \ \alpha_i + \beta_i = 1$. For convenience, we denote $y_i^{\text{real}}, y_i^{\text{pseudo}}, \mathcal{F}(x_i, \theta)$ using $y_i^r, y_i^p, p_i$ respectively.

$$\alpha_i \mathcal{L}(p_i, y_i^r) + \beta_i \mathcal{L}(p_i, y_i^p) = \sum_{l=1}^{L} \alpha_i y_{i,l}^r \log p_{i,l} \tag{15}$$

$$+ \beta_i y_{i,l}^p \log p_{i,l} = \sum_{l=1}^{L} (\alpha_i y_{i,l}^r + \beta_i y_{i,l}^p) \log p_{i,l} \tag{16}$$

Due to that $\mathcal{L}(\cdot)$ is the cross-entropy loss, we have $\sum_{l=1}^{L} y_{i,l}^r = \sum_{l=1}^{L} y_{i,l}^p = 1$. Then $\sum_{l=1}^{L} \alpha_i y_{i,l}^r + \beta_i y_{i,l}^p = \alpha_i + \beta_i$. So if $\alpha_i + \beta_i = 1$, we have

$$\sum_{l=1}^{L} (\alpha_i y_{i,l}^r + \beta_i y_{i,l}^p) \log p_{i,l} = \mathcal{L}(p_i, \alpha_i y_i^r + \beta_i y_i^p) \tag{17}$$

$$= \mathcal{L}(p_i, (1 - \beta_i) y_i^r + \beta_i y_i^p) \tag{18}$$

### B.2 GRADIENT USED FOR UPDATING $\theta$

We derivative the update rule for $\boldsymbol{\alpha}, \boldsymbol{\beta}$ in Eq. 10.

$$\alpha_{t,i} = -\eta \frac{\partial}{\partial \alpha_i} (\sum_{j=1}^{m} f_j^v(\hat{\theta}_{t+1}))\Big|_{\alpha_i=0} \tag{19}$$

$$= -\eta \sum_{j=1}^{m} \nabla f_j^v(\hat{\theta}_{t+1})^T \frac{\partial \hat{\theta}_{t+1}}{\partial \alpha_i}\Big|_{\alpha_i=0} \tag{20}$$

$$= -\eta \sum_{j=1}^{m} \nabla f_j^v(\hat{\theta}_{t+1})^T \tag{21}$$

$$\frac{\partial(\theta_t - \lambda \nabla(\sum_k \alpha_k \ f_k(\theta) + \beta_k \ g_k(\theta))\big|_{\theta=\theta_t})}{\partial \alpha_i}\Big|_{\alpha_i=0} \tag{22}$$

$$= \eta \lambda \sum_{j=1}^{m} \nabla f_j^v(\theta_t)^T \nabla f_i(\theta_t) \tag{23}$$

$$\beta_{t,i} = -\eta \frac{\partial}{\partial \beta_i} (\sum_{j=1}^{m} f_j^v(\hat{\theta}_{t+1}))\Big|_{\beta_i=0} \tag{24}$$

$$= -\eta \sum_{j=1}^{m} \nabla f_j^v(\hat{\theta}_{t+1})^T \frac{\partial \hat{\theta}_{t+1}}{\partial \beta_i}\Big|_{\beta_i=0} \tag{25}$$

$$= -\eta \sum_{j=1}^{m} \nabla f_j^v(\hat{\theta}_{t+1})^T \tag{26}$$

$$\frac{\partial(\theta_t - \lambda \nabla(\sum_k \alpha_k \; g_k(\theta) + \beta_k \; g_k(\theta))\Big|_{\theta=\theta_t})}{\partial \beta_i}\Bigg|_{\beta_i=0} \tag{27}$$

$$= \eta \lambda \sum_{j=1}^{m} \nabla f_j^v(\theta_t)^T \nabla g_i(\theta_t) \tag{28}$$

Then $\theta_{t+1}$ can be calculated by Eq. 10 using the updated $\alpha_{t,i}, \beta_{t,i}$.

### B.3 CONVERGENCE

This section provides the proof for covergence (Theorem 1)

**Theorem.** *Suppose that the training loss function $f, g$ have $\sigma$-bounded gradients and the validation loss $f^v$ is Lipschitz smooth with constant L. With a small enough learning rate $\lambda$, the validation loss monotonically decreases for any training batch B, namely,*

$$G(\theta_{t+1}) \leq G(\theta_t), \tag{29}$$

*where $\theta_{t+1}$ is obtained using Eq. 10 and G is the validation loss*

$$G(\theta) = \frac{1}{M} \sum_{i=1}^{M} f_i^v(\theta), \tag{30}$$

*Furthermore, Eq. 29 holds for all possible training batches only when the gradient of validation loss function becomes 0 at some step t, namely, $G(\theta_{t+1}) = G(\theta_t) \; \forall B \Leftrightarrow \nabla G(\theta_t) = 0$*

*Proof.* At each training step $t$, we pick a mini-batch $B$ from the union of training and validation data with $|B| = n$. From section B we can derivative $\theta_{t+1}$ as follows:

$$\theta_{t+1} = \theta_t - \lambda \sum_{i=1}^{n} (\alpha_{t,i} \nabla f_i(\theta_t) + \beta_{t,i} \nabla g_i(\theta_t)) \tag{31}$$

$$= \theta_t - \eta \lambda^2 M \sum_{i=1}^{n} (\nabla G^T \nabla f_i \nabla f_i + \nabla G^T \nabla g_i \nabla g_i) \tag{32}$$

We omit $\theta_t$ after every function for briefness and set $m$ in section B equals to $M$. Since $G(\theta)$ is Lipschitz-smooth, we have

$$G(\theta_{t+1}) \leq G(\theta_t) + \nabla G^T \Delta\theta + \frac{L}{2} ||\Delta\theta||^2. \tag{33}$$

Then we show $\nabla G^T \Delta\theta + \frac{L}{2} ||\Delta\theta||^2 \leq 0$ with a small enough $\lambda$. Specifically,

$$\nabla G^T \Delta\theta = -\eta \lambda^2 M \sum_i (\nabla G^T \nabla f_i)^2 + (\nabla G^T \nabla g_i)^2. \tag{34}$$

Then since $f_i, g_i$ have $\sigma$-bounded gradients, we have

$$\frac{L}{2}||\Delta\theta||^2 \leq \frac{L\eta^2\lambda^4 M^2}{2}\sum_i (\nabla G^T\nabla f_i)^2||\nabla f_i||^2 \tag{35}$$

$$+ (\nabla G^T\nabla g_i)^2||\nabla g_i||^2 \tag{36}$$

$$\leq \frac{L\eta^2\lambda^4 M^2\sigma^2}{2}\sum_i (\nabla G^T\nabla f_i)^2 + (\nabla G^T\nabla g_i)^2 \tag{37}$$

Then if $\lambda^2 < \frac{2}{\eta\sigma^2 ML}$,

$$\nabla G^T\Delta\theta + \frac{L}{2}||\Delta\theta||^2 \leq (\frac{L\eta^2\lambda^4 M^2\sigma^2}{2} - \eta\lambda^2 M) \tag{38}$$

$$\sum_i (\nabla G^T\nabla f_i)^2 + (\nabla G^T\nabla g_i)^2 \leq 0. \tag{39}$$

Finally we prove $G(\theta_{t+1}) = G(\theta_t)\ \forall B \Leftrightarrow \nabla G(\theta_t) = 0$: If $\nabla G(\theta_t) = 0$, from section B we have $\alpha_{t,i} = \beta_{t,i} = 0$, then $\theta_{t+1} = \theta_t$ and thus $G(\theta_{t+1}) = G(\theta_t)\ \forall B$. Otherwise, if $\nabla G(\theta_t) \neq 0$, we have

$$0 < ||\nabla G||^2 = \nabla G^T\nabla G = \frac{1}{M}\sum_{i=1}^{M}\nabla G^T\nabla f_i^v, \tag{40}$$

which means there exists a $k$ such that $\nabla G^T\nabla f_k^v > 0$. So for the mini-batch $B_k$ that contains this example, we have

$$G(\theta_{t+1}) - G(\theta_t) \leq \nabla G^T\Delta\theta + \frac{L}{2}||\Delta\theta||^2 \tag{41}$$

$$\leq (\frac{L\eta^2\lambda^4 M^2\sigma^2}{2} - \eta\lambda^2 M) \tag{42}$$

$$\sum_{i\in B}(\nabla G^T\nabla f_i)^2 + (\nabla G^T\nabla g_i)^2 \tag{43}$$

$$\leq (\frac{L\eta^2\lambda^4 M^2\sigma^2}{2} - \eta\lambda^2 M)\nabla G^T\nabla f_k^v \tag{44}$$

$$< 0. \tag{45}$$

