# OpenReview forum: "L2B: Learning to Bootstrap Robust Models for Combating Label Noise"
_ICLR.cc/2024/Conference — ICLR 2024 Conference Withdrawn Submission_

### Official Review · Reviewer_Pagb · 2023-10-16

**Soundness:** 3 good
**Presentation:** 3 good
**Contribution:** 3 good
**Rating:** 6
**Confidence:** 4

**Summary:**

This paper proposes a novel method, namely Learning to Bootstrap (L2B), which leverages a joint reweighting mechanism to train models using their own predictions to bootstrap themselves, where the adaptable weights are obtained by meta learning. Extensive experiments demonstrate the superior performance of the proposed method over recent state-of-the-art methods.

**Strengths:**

1. The paper is well written.
2. This idea is very clear, simple and effective.
3. The proposed method has very good performance.
4. The proposed method is easy to reproduce.

**Weaknesses:**

1.  The contributions are not very clear. It is better to summarize the contribution.
2.  It is not clear why \alpha+\beta \neq 1 can significantly boost the model performance?
3.  There is no latest comparative methods, like the methods in Tables 3-4, the latest one is from 2021. It is better to add more latest comparative methods.
4. Some sentences are too long and different to understand. For example, "we propose a  novel machine learning method called Learning to Bootstrap (L2B) that lever-
ages a joint reweighting mechanism to train models using their own predictions to bootstrap themselves without being adversely affected by erroneous pseudo-labels". Please short them.

**Questions:**

1. Why \alpha+\beta \neq 1 can significantly boost the model performance? Please provide more analysis.
2. What are the limitations of the proposed method, and please point out the future work.

---

### Official Review · Reviewer_iafb · 2023-10-30

**Soundness:** 3 good
**Presentation:** 2 fair
**Contribution:** 2 fair
**Rating:** 3
**Confidence:** 5

**Summary:**

This paper studies to improve label correction method. Specifically, they try to achieve adaptive instance and label reweighting based on bootstrapping loss method in a meta-learning manner. This method is expected to more effectively exploits noisy examples compared with sample reweighting method.

**Strengths:**

- Give a more general loss formulation for bootstrapping loss.
- Learning the new loss in a meta-learning manner, and achieve a better performance compared with original bootstrapping loss and some sample weighting methods using meta learning.

**Weaknesses:**

- The idea is not novel. To improve the bootstrapping loss, [A] proposes dynamic hard and soft bootstrapping losses by individually weighting each samples. The sample-wise weights mean that the sample whether or not belongs to clean labels. The clean samples rely on their ground-truth labels, while noisy ones let their loss being dominated by their class predict. To determine the weights, [A] use two-component Beta Mixture Model (BMM), and [B] use two-component GMM to fit the max-normalized loss. For meta learning methods, [C] uses learning to reweight techniques and [D] use meta-weight-net techniques to learn weights. As compared, the used techniques in this paper is not novel.
[A] Arazo, E., Ortego, D., Albert, P., O’Connor, N., McGuinness, K.: Unsupervised label noise modeling and loss correction. In ICML, 2020.
[B] Li, J., Socher, R., Hoi, S.C.: Dividemix: Learning with noisy labels as semi-supervised learning. In ICLR, 2020.
[C] Zhang, Z., Zhang, H., Arik, S.O., Lee, H., Pfister, T.: Distilling effective supervision from severe label noise. In CVPR, 2020.
[D] Shu J, Yuan X, Meng D, et al. Cmw-net: Learning a class-aware sample weighting mapping for robust deep learning[J]. IEEE Transactions on Pattern Analysis and Machine Intelligence, 2023.

- Though this paper uses two group weights to bring more freedom to the bootstrapping loss, the motivation is not clear to readers. Though Fig.1(a) shows that original bootstrapping loss is sensitive to the reweighting parameter, it is also suitable for [A-D], since they also deal with such problem. Unfortunately, it is not clear why authors use such more general loss function. Page 3 says it "treats all examples as equally important during training, which could easily cause overfitting for biased training data." This claim is very subjective, and there exist no evidence to support it. Page 4 says "but rather conducts implicit relabeling during training by reweighting different loss terms", which is not clear to readers. And I can not find why this bring benifits for the bootstrapping loss.

- Page 4 says "Note that this form is also similar to self-distillation". I have checked the form therein. Its form is the convex combination weights in [A,B,C,D]. Thus the folowing claim is not convincing, since [C,D] also use validation set to learn weights in a meta-learning manner.


- The display in Fig.4 can not demostrate the effectiveness of proposed method: the weights relationship of the last two columns is not reasonable. When the pseudo label is equal to the noisy label, they have identical weights. If do this, it can not bring benifits for learning, since the learned weights make it overfit the noisy data. This can not explain the effectiveness of proposed method.

**Questions:**

- The novelty of the proposed method. The experimental validation can not support the effectiveness of proposed method. Especically, how such weighting learning help improve learning is not clear. The display in Fig.3 can not support the effect weighting learning. When the pseudo label is equal to the noisy label, the learned weights push the learning towards the direction of overfitting noisy data.
- The technical novelty is limited, since the algorithm and theoretical convergence guarantees used in this paper are already proposed in L2RW [1] and Meta-Weight-Net [2].


[1] Learning to reweight examples for robust deep learning
[2] Meta-weight-net: Learning an explicit mapping for sample weighting

---

### Official Review · Reviewer_Ss7A · 2023-10-31

**Soundness:** 3 good
**Presentation:** 3 good
**Contribution:** 2 fair
**Rating:** 5
**Confidence:** 4

**Summary:**

The paper presents the L2B method, a new approach for dynamic loss weight assignment for both true label samples and pseudo label samples. The authors trained a first-order meta-learner to set the weights based on a clean validation set. This method was tested on three common datasets: CIFAR, Clothing-1M, and PROMISE12, and achieved good performance.

**Strengths:**

1. This paper is generally well-organized and easy to follow.
2. The idea of separately learning loss weights for the true-label term and the pseudo-label term is rational and logical for tasks involving learning with noisy labels.
3. The results shown on three datasets validate the effectiveness of the proposed method.

**Weaknesses:**

1. The motivation clarification of L2B could benefit from further improvement. While the paper states, "In contrast to prior works that individually reweight labels or instances, our paper introduces a novel approach to concurrently adjust both, elegantly unified under a meta-learning framework. We term our method as Learning to Bootstrap (L2B), as our goal is to enable the network to self-boost its capabilities by harnessing its own predictions in combating label noise," a clearer, high-level or theoretical explanation might enhance the portrayal of the paper's innovation.

2. L2B can potentially serve as a plug-in module for LNL methods. It would be beneficial to see additional results, especially when L2B is integrated with state-of-the-art (SOTA) methods, to further test the proposed method's effectiveness.

3. The related work section appears dated. There's a noticeable lack of references from 2023, with most of the cited works being from before 2022 and only three from 2022. It raises the question of whether any recent work from 2023 might have similar ideas. The authors might undertake a more meticulous survey and comparison with the latest SOTA methodologies.

**Questions:**

1. In the experimental section, the reviewer wonders how the method would handle the absence of clean validation, such as with datasets like CIFAR-N or WebVision. How would such scenarios be addressed?

2. L2B employs a first-order meta-learning algorithm to study two parameters. Have the authors considered trying a second-order method similar to MAML? Would the performance see any improvements?

3. The reviewer notes that within L2B, the weights assigned to the clean-label term seem to be the same as the weights for both the pseudo sample and label. Have the authors thought about using different weights for samples and labels? Adopting such a strategy might appear more rational.

---

### Official Review · Reviewer_Qjb7 · 2023-11-01

**Soundness:** 3 good
**Presentation:** 3 good
**Contribution:** 3 good
**Rating:** 6
**Confidence:** 2

**Summary:**

This paper introduces L2B, a new meta learning-based technique for learning from noisy labels that involves training on predictions from the model, while adjusting importance weighting of pseudolabels. Additionally, L2B can be combined with existing techniques for learning from noisy labels. Theoretically, the authors show that their method reduces to the existing bootstrapping loss, and empirically, they show that their method performs well on several benchmark datasets.

**Strengths:**

- The authors perform extensive experimentation on synthetic datasets, one dataset with real label noise, and compare to a large number of methods.
- The convergence analysis is a great contribution to the paper.
- The paper is well-written. Before getting to the experiments section, I was curious about whether or not the means of simplex projection for \alpha and \beta mattered in the method, since the authors perform an approximation. The authors seem to have pre-empted this potential question of mine, as they include an ablation that evaluates using the softmax of \alpha and \beta.

**Weaknesses:**

While the authors evaluate on one dataset with real label noise, Clothing-1M, there are a variety of other real-world settings where coping with (potentially systematic) label noise is important. It would be great to include additional real-world settings that may have systematic label noise. For example, label noise is a problem that is often inherent to the family of programmatic weak supervision techniques.

**Questions:**

- Given Euqation 9, it's unclear to me if \alpha and \beta actually sum up to 1 in the final algorithm. Can the authors clarify this? In earlier discussion of the algorithm, it seems as though allowing \alpha and \beta to not sum to 1 is crucial for its final performance.
- The ablation studies on \alpha and \beta are interesting and a useful contribution to the paper. While I imagine that performing an exact simplex projection using one of the algorithms for doing so would be slow (mainly because it must be done separately for each sample), have the authors tried other methods besides the softmax or the sigmoid?
- In the current formulation that does not use the sigmoid or softmax, parameters are allowed to be zero. Is it common for either or both of them to be zero? Could this by why projecting to the interior of the simplex via softmax or sigmoid improves the stability of training?
- It is not clear to me whether or not a validation set is required. It is mentioned in several places in the paper that a validation set is not required, however, in 3.1, "In this work, we also assume that there is a small unbiased and clean validation set." Can the authors clarify this point?